# JC Virus Seroprevalence and JCVAb Index in Polish Multiple Sclerosis Patients Treated with Immunomodulating or Immunosuppressive Therapies

**DOI:** 10.3390/jcm10091998

**Published:** 2021-05-06

**Authors:** Robert Bonek, Wojciech Guenter, Robert Jałowiński, Anna Karbicka, Anna Litwin, Maciej Maciejowski, Radosław Zajdel, Karolina Zajdel, Veronique Petit, Konrad Rejdak

**Affiliations:** 1Department of Neurology and Clinical Neuroimmunology, Regional Specialist Hospital, 86-300 Grudziadz, Poland; 2Foundation Supporting Development of Neurology and Clinical Neuroimmunology MoA, 85-654 Bydgoszcz, Poland; 3Department of Clinical Neuropsychology, Nicolaus Copernicus University, 87-100 Torun, Poland, and Collegium Medicum, 85-094 Bydgoszcz, Poland; wojciech.guenter@gmail.com; 4Department of Neurology, Regional Hospital, 71-455 Szczecin, Poland; rojal@esculap.pl (R.J.); anuszek-4@o2.pl (A.K.); 5Department of Neurology, Regional Hospital, 10-561 Olsztyn, Poland; annalitw@gmail.com; 6KMK Clinical, MS Center, 40-571 Katowice, Poland; maciejmaciejowski@poczta.onet.pl; 7Chair of Business Informatics, University of Lodz, 90-214 Lodz, Poland; radoslaw.zajdel@uni.lodz.pl; 8Department of Medical Informatics and Statistics, Medical University of Lodz, 90-645 Lodz, Poland; karolina@interforum.pl; 9Department of Neurology, Medical University of Lublin, 20-090 Lublin, Poland; vpetit@onet.pl (V.P.); konradrejdak@umlub.pl (K.R.)

**Keywords:** multiple sclerosis, John Cunningham virus, anti-JCV antibody, antibody index, disease-modifying therapies

## Abstract

The use of a highly-effective treatment for multiple sclerosis (MS) is associated with a severe risk of developing complications, such as progressive multifocal leukoencephalopathy (PML) caused by the John Cunningham virus (JCV). The aim of this study was to evaluate the correlation between anti-JCV Ab seroprevalence, anti-JCV AI, demographic and clinical factors as well as the type of therapy used in the Polish MS population. This is a multicentre, prospective and cross-sectional study involving 1405 MS patients. The seroprevalence of anti-JCV Ab and anti-JCV AI levels as well as AI categories were analysed with the use of a second-generation two-step ELISA test (STRATIFY JCV DxSelect). The overall prevalence of anti-JCV Ab was 65.8%. It was shown that seroprevalence increases with the patient’s age. The seroprevalence was significantly associated with the treatment type, and the highest values (76%) were obtained from immunosuppressant-treated patients. Overall, 63.3% of seropositive patients had an antibody index (AI) level of >1.5. In the seropositive patient group, the mean AI level amounted to 2.09. Similarly to the seroprevalence, AI levels correlated with the patient’s age; AI level for patients above 40 years old and from subsequent age quintiles plateaued, amounting to at least 1.55. Patients treated with immunosuppressants and immunomodulatory drugs obtained the highest (1.67) and lowest (1.35) AI levels, respectively. Of the immunosuppressants used, the highest mean AI levels were observed in mitoxantrone and cladribine groups, amounting to 1.75 and 1.69, respectively. In patients treated with immunomodulatory drugs, the lowest AI levels were observed in the dimethyl fumarate (DMF) group (1.11). The seroprevalence rate in the Polish MS population is one of the highest in Europe. The majority of seropositive patients had an anti-JCV Ab level qualifying them for a high-risk category. The highest mean AI levels are observed in patients receiving immunosuppressants, especially mitoxantrone and cladribine. Patients receiving immunomodulatory drugs have lower AI levels compared to treatment-naïve subjects, especially when treated with DMF. Further studies, especially longitudinal studies, are required to determine the impact of MS drugs on the seroprevalence of anti-JCV Ab and AI levels.

## 1. Introduction

The John Cunningham virus (JCV) is an aetiological factor responsible for the development of a rare, opportunistic and often fatal, demyelinating disease of the central nervous system (CNS) called progressive multifocal leukoencephalopathy (PML). The JC virus is a common infection that is widely present in the general population. In healthy adults, JCV antibodies (JCV Ab) are observed in 33% to 91% of the population [1]. The majority of persons acquire the infection during childhood without evident pathological or clinical consequences. In the latent form, the virus remains in many tissues, such as the kidneys, tonsils, peripheral blood leukocytes, including B cells, and brain, for a duration of many years. The mechanism of latent virus reactivation and PML development is not fully explained. The latent form of JC virus in the brain can be reactivated as a result of the body response to extracellular cytokines initiating viral replication due to an immunosuppression-related lack of immunological surveillance [2,3,4,5].

Advances in the therapy for autoimmune diseases—initially with classical immunosuppressants, such as azathioprine, chlorambucil, cyclophosphamide, and methotrexate, and recently with monoclonal antibodies (mAb), such as efalizumab, natalizumab, rituximab or ocrelizumab—increased the rate of treatment-related PML [1,6,7].

In recent years, the occurrence of PML in patients with multiple sclerosis (PwMS) resulting from the use of highly-effective biological treatment gained particular importance. The first cases of PML related to the treatment with natalizumab (NTZ), a mAb targeting α4β1 and α4β7 integrins, were reported almost 15 years ago in phase 3 clinical trials evaluating patients with relapsing–remitting multiple sclerosis (RRMS). In the following years, the incidence of PML in PwMS treated with NTZ increased. Further, a detailed risk stratification tool was developed for this treatment. Initially, three risk factors associated with natalizumab-related PML occurrence were determined: the presence of anti-JCV Ab in blood serum, the use of an immunosuppressive treatment prior to NTZ and a duration of natalizumab therapy of more than 2 years. The results of studies presented in the following years allowed for the identification of one more risk factor in seropositive patients, which is an anti-JCV Ab index (AI) level of >1.5. However, PML cases reported in RRMS patients treated with fingolimod (FGL) or dimethyl fumarate (DMF), and in RRMS and PPMS patients receiving ocrelizumab, showed that the above-mentioned clinical problem is quite complex [1,8,9,10,11].

The existing data about the serostatus of the MS population have been mainly derived from studies assessing the prevalence of anti-JCV Ab in PwMS from Europe, North America, the Middle East and East Asia. In MS patients, the seropositivity rate worldwide is approximately 57% and was determined based on the previous data originating from two international studies [12,13]. There were initial suggestions that the seroprevalence of anti-JC Ab does not depend on the specific geographical pattern. However, regardless of those suggestions, it seems that such differentiation exists. The latest studies from Asia showed a significantly higher anti-JCV Ab seropositivity rate in Chinese, Japanese and South Korean populations, ranging from 69.5% to 80% [14,15,16]. The analysis of anti-JCV Ab index levels also shows discrepancies with the highest values observed in East Asia.

In PwMS, according to the current state of knowledge, the seroprevalence and seropositivity are mainly linked with the patient’s age at the time of sample draw. The results of studies evaluating the correlation with other demographic or clinical factors were inconsistent or did not show any relation with an anti-JCV Ab serostatus. It also concerned the use of disease-modifying therapy (DMT) [12,13,14,15,16,17,18,19,20]. Regarding the fact that prior immunosuppressive treatment plays a crucial role as a risk factor for PML development in patients receiving NTZ [8,9], it seems that the identification of eventual seroprevalence differences in treatment-naive subjects as well as in patients receiving immunomodulatory drugs or immunosuppressants is of crucial importance.

This study aimed to determine the seroprevalence of anti-JCV Ab in the Polish population of patients receiving immunomodulatory, immunosuppressive drugs as well as in treatment-naïve persons. This study also aimed to define the correlation between the serostatus, anti-JCV Ab index level and demographic and clinical factors as well as the DMT used.

## 2. Materials and Methods

### 2.1. Patients

This was a multicentre, prospective and cross-sectional study conducted at five sites managing the diagnostics and treatment of multiple sclerosis located in five regions of Poland (Bydgoszcz/Grudziadz, Katowice, Lublin, Olsztyn and Szczecin). Data were collected from July 2014 to January 2018. This study enrolled 1405 patients with MS diagnosed with the use of the 2010 modified McDonald criteria [21]. 

The participants had to be over 18 years old. The following demographic and clinical data were collected: age, sex, disease duration (from the onset of the first symptoms), disability level measured with the EDSS score, MS course, type of treatment used (immunomodulatory, immunosuppressive or treatment-naive) and drugs used (interferon beta, glatiramer acetate, dimethyl fumarate, fingolimod, natalizumab, cyclophosphamide, mitoxantrone, and cladribine in vials for subcutaneous administration). The exclusion criteria included age below 18 years; therapy with corticosteroids within four weeks before serum sampling for the measurement of the anti-JCV Ab level; the use of intravenous immunoglobulins within the previous six months.

The primary endpoint was the prevalence of anti-JCV Ab in MS patients and the correlation between the presence of anti-JCV Ab and the collected demographic and clinical data. The secondary endpoints included an anti-JCV Ab index (AI) level measurement and the correlation between the AI level categories as well as demographic and clinical data.

### 2.2. Samples

Anti-JCV Ab serostatus and anti-JCV Ab index testing was performed after the start of the treatment in all patients. There was no constant time interval between the treatment initiation and the testing, but this interval was not less than 3 months.

All the samples tested for the anti-JCV Ab serostatus and index were analysed by the reference laboratory (UNILABS) located in Copenhagen (Denmark). The second-generation confirmatory ELISA (STRATIFY JCV™ DxSelect–STRAFITY2) test was used for testing the sera for anti-JCV antibodies and index levels. The testing procedure consisted of a screening enzyme-linked immunosorbent assay (ELISA) and a (supplemental) confirmatory test. During the screening, the measured anti-JCV Ab index levels amounting to <0.2 and >0.4 were considered negative and positive, respectively. Samples with index levels between 0.2 and 0.4 underwent a confirmatory test (second step), in which the results of >45% were classified as anti-JCV Ab positive [22].

### 2.3. Statistical Analysis

The results were analysed with the use of statistical methods, including some multidimensional tests. The Shapiro–Wilk test was used to assess normal distribution. The studied characteristics of non-normal distribution as well as qualitative and quantitative data were analysed with the use of non-parametric tests, including the Kruskal–Wallis ANOVA, Pearson’s chi-squared and Mann–Whitney U tests. General descriptive statistics methods were also used. A logistic regression model was used for multivariate analysis, where applicable. Analysis of covariance (ANCOVA) was used to control for the effect of some continuous variables that were not of primary interest when evaluating the effect of categorical independent variables on dependent one. The statistically significant *p* level was at <0.05.

## 3. Results

### 3.1. Patients

The clinical and demographic data of MS patients are summarised in Table 1.

### 3.2. Prevalence and Index of Anti-JCV Ab

Anti-JCV Ab was detected in 924 of the 1405 patients enrolled in the study group. This means that the overall prevalence index of anti-JCV Ab amounted to 65.8%. In the studied PwMS cohort, no significant differences in the anti-JCV Ab seroprevalence by patients’ sex (66.1% in females and 65.1% in males) were observed. The analysis of anti-JCV Ab seroprevalence considering the MS clinical course showed higher seropositivity in SPMS (68.3%) and PPMS (75.3%) compared to RRMS (63.2%) patients; the difference between RRMS and PPMS was significant. The seropositivity rate increased along with the increasing disability level measured on the EDSS scale (Table 2).

After an additional request, the anti-JCV Ab index levels were obtained for the entire cohort enrolled in this study, including both seropositive and seronegative patients. Overall, the mean anti-JCV Ab index (AI) amounted to 1.44 ± 1.29 (median 0.95); the lowest and highest levels were 0.03 and 4.51, respectively. In the seropositive patient group, the mean AI level was 2.09 (median 2.2) with a range of 0.2–4.51. In the seronegative patient group, the above-mentioned parameters were as follows: AI 0.2 (median 0.19), range 0.03–0.4. The analysis of the correlation between the anti-JCV Ab index level and the patient’s sex and disability level did not show any significant differences; however, we observed the trend towards the AI level increase along with the increasing disability level. The correlation between the anti-JCV Ab index (AI) and clinical disease course showed that patients with RRMS have the lowest (1.39) while patients with PPMS have the highest (1.71) AI levels; the observed difference was statistically significant (Table 2).

The analysis of the results concerning a correlation between the patient’s age and JCV serostatus showed a robust relationship between the increasing anti-JCV Ab seropositivity and the increasing patients’ age at the time of the sample draw (*p* < 0.0001; Table 3). The seropositivity was increasing from 52.4% in the youngest to 79.6% in the oldest group of patients. We also observed the relation between the serostatus and disease duration; the highest anti-JCV Ab prevalence was observed in patients with the longest MS duration (Table 3).

We have also noted a significant increase in the anti-JCV AI level that occurred along with the increasing age of PwMS from 1.15 ± 1.24 in a group aged 18–29 years to 1.58 ± 1.30 in a group aged 50–59 years (Table 3; Figure 1); without the impact of disease duration on the AI level (Table 3). Moreover, in patients from the age quintiles above 40 years, the AI value plateaued.

The results presented above were used to conduct a covariance analysis and an analysis of the impact of the patient’s age on the remaining correlations studied. The ANCOVA model included the course of MS, EDSS score and disease duration as quality-independent variables and age as a continuity-independent variable. The results showed that only an older age at the time of serum sampling is significantly correlated with an increase in the prevalence of anti-JCV Ab in the study cohort. At the same time, the quality characteristics (MS course, EDSS score and disease duration) do not significantly impact anti-JCV Ab serostatus and AI (*p* > 0.05).

### 3.3. Disease-Modifying Therapy and Anti-JCV Ab Serostatus and Index

Table 4 shows the prevalence of the anti-JCV Ab and anti-JCV AI levels depending on the treatment used and in treatment-naïve patients. In our cohort, the highest seroprevalence was observed in patients treated with classical immunosuppressants (76%). Serostatus results obtained in the treatment-naïve cohort (65.3%) and patients receiving immunomodulatory drugs and selective immunosuppressants (62.4%) were similar, and the differences between IS and IM groups were statistically significant. Of all the evaluated drugs, the highest anti-JCV Ab prevalence was observed in the cyclophosphamide group (78.8%). However, the difference between that group and the patients receiving other immunosuppressants, i.e., mitoxantrone or cladribine in vials for subcutaneous (SC) administration was not significant.

Regarding immunomodulatory drugs, the lowest anti-JCV Ab prevalence was observed in patients treated with DMF (45.5%) followed by IFN (63.8%), while the highest seroprevalence was obtained patients receiving GA (68.2%). However, the in-between group differences were insignificant. The serostatus results obtained for fingolimod and natalizumab are linked to the recommendations on the qualification of patients to treatment with those agents. NTZ is mainly used in seronegative, while FGL in seropositive and to a lesser extent in seronegative patients.

Similarly to the seroprevalence, the highest mean anti-JCV AI level (1.67) and median (1.54) were observed in patients treated with immunosuppressants compared to the treatment-naïve cohort (mean and median of 1.44 and 0.93, respectively) and PwMS receiving immunomodulatory drugs and selective immunosuppressants (1.35 and 0.74, respectively). The in-between group differences were statistically significant. Regarding classical immunosuppressants, the highest mean anti-JCV AI levels were observed in the mitoxantrone (1.75) group, followed by cladribine (1.69) and cyclophosphamide (1.50) but without a statistically significant difference. In addition, the analysis of the anti-JCV AI levels in patients receiving immunomodulatory drugs did not show any statistically significant differences. The same factor as that described for the seroprevalence could affect the AI levels measured in the FGL and NTZ subgroups.

### 3.4. Anti-JCV Ab Index Categories

Table 5 and Table 6 show the anti-JCV AI level categories for the entire study cohort and the particular demographic and clinical parameters. In a group of 924 seropositive patients, 22 (2.4%) persons had an AI between 0.2 and 0.4; 188 (20.3%) had an AI from 0.4 to 0.9; 129 (14%) from 0.9 to 1.5; and 585 (63.3%) patients achieved an AI of >1.5. We also verified the proportion of patients with very high index levels > 3.0; such an AI level was detected in 260 patients, representing 18.5% of the entire study cohort and 28.1% of seropositive patients. No significant differences concerning the AI level categories by the analysed demographic and clinical parameters were observed. Figure 2 shows the distribution of anti-JCV AI levels by the patient’s age.

Table 7 shows the anti-JCV AI categories by the disease-modifying therapy (DMT) used, the use of particular drugs, and in the treatment-naïve subgroup. The analysis of treatment-naïve, IS- and IM-treated subgroups demonstrated that for AI level category > 1.5, the highest prevalence is observed in patients receiving classical immunosuppressants (51.4%). The lowest prevalence is observed in patients treated with immunomodulatory drugs and selective immunosuppressants (38.3%); however, the difference is statistically significant. Regarding IS treatment, AI levels > 1.5 were observed in 58.2% treated with mitoxantrone, and in 46.4% and 40.4% of PwMS receiving cladribine and cyclophosphamide, respectively. However, the identified differences were insignificant. In addition, no statistically significant differences in the prevalence of anti-JCV AI categories were observed in patients receiving immunomodulatory agents, such as GA, IFNβ or DMF.

## 4. Discussion

The existing data on the seroprevalence of anti-JCV Ab and anti-JCV AI levels in the population of MS patients mainly originate from trials in which PwMS were receiving disease-modifying therapy (DMT), especially natalizumab [23,24,25,26,27,28]. The vital characteristic of our study cohort comprising MS patients is the fact that nearly one half of them were treatment naïve, while the remaining study population was receiving DMT with the use of immunomodulatory or immunosuppressive agents; the proportion of patients treated with natalizumab was minimal. 

More interestingly, this is the largest trial evaluating the JCV seroprevalence in the PwMS population with the use of the second-generation STRATIFY JCV™ DxSelect test—STRAFITY2. Our trial had a prospective and cross-sectional design. Some unique aspects of thid study involved more numerous study groups of patients with progressive MS forms (PPMS and SPMS) and, most importantly, more numerous cohorts of patients receiving immunosuppressive therapy. Importantly, this is also the first study comparing and evaluating the anti-JCV AI levels by the treatment used (i.e., comparing patients treated with immunosuppressive, immunomodulatory agents as well as a treatment-naïve subgroup) and analysing the AI levels for particular MS drugs to such a large extent. 

In our multicentre study, the overall seroprevalence of anti-JCV Ab in the Polish population of MS patients was 65.8%. Our data were obtained with the use of the STRATIFY JCV™ DxSelect test (second-generation test), which compared to the first-generation test STRATIFY JCV™ provides a higher ability to detect anti-JCV Ab in cases of low response and higher reproducibility [22]. This may potentially cause some limitations when comparing our data with the results obtained in MS populations treated in other countries; especially with the results obtained from the previous trials, where the STRATIFY JCV™ test was used. Nevertheless, it seems that the overall results obtained with the use of the first- and second-generation tests are consistent. Of course, we should remember that the use of the second-generation test resulted in higher JCV seroprevalence rates [13,17,19,20]. 

We compared our results concerning the overall anti-JCV Ab seroprevalence with data derived from two extensive international studies using the first-generation test to evaluate JCV serostatus (in which the overall worldwide JCV seropositivity amounted to 57.1% and 57.6%). Based on that comparison, we noticed that anti-JCV Ab prevalence in the Polish population is markedly higher compared to the global mean value and one of the highest versus previous studies using the STRATIFY JCV™ test [12,13]. However, when comparing our results with data derived from other European countries (where STRATIFY JCV™ DxSelect test was used), we can state that anti-JCV Ab prevalence in the Polish cohort of PwMS is higher than in the Spanish (62.3% and 60.5%), Czech (59%), British (59%), Finnish (57.4%) or French (49.7%) populations [17,18,19,27,28,29]. In comparison with the Austrian (72.1% and 52.3%) and Portuguese (68.2% and 60.8%) data, the seroprevalence in the Polish population is lower or higher. However, it is noteworthy that both Austrian cohorts were less numerous, while the Portuguese study demonstrating higher seroprevalence had a single-centre design [20,30,31,32]. The serostatus discrepancies between different European countries seem to be difficult to explain; perhaps it is a matter of a well-selected study cohort or its size.

The fact that the data were obtained in other regions, such as the countries of North America, the Middle East or Australia and Brazil [13,23,33,34,35], is consistent with the European results and, therefore, Poland can be classified as a region of high JCV prevalence in the population of MS patients.

The countries of East Asia (Japan, China, South Korea) markedly differ from other regions because they show JCV seropositivity rates ranging from 69.5% to 80% [14,15,16]. The reason for such a large discrepancy in the seroprevalence of anti-JCV Ab between countries in East Asia and the rest of the world (including our data) remains unclear. The recently presented hypotheses aim to explain the phenomenon mentioned above by the differences in the prevalence of specific JCV genotypes depending on the geographic region. It was demonstrated that of the four main JCV genotypes, type 1 dominates in Europe and type 2 in Asia. Therefore, complex relationships resulting from the impact of genetic and immunological factors on the interaction between the specific virus type and the host may explain the observed differences in the seroprevalence rate [15].

Data from the previous studies showed the link between an increase in the anti-JCV Ab seropositivity rate and patients’ age, both with the use of the first- [12,13,23,25,26,36] and second-generation tests [17,18,19,20,27,33]. Our data supported the above correlation indicating an increase in the JCV seropositivity rate along with the patient’s age. This proves the thesis on the continuous increase in JCV infections with age and confirms the highest anti-JCV Ab prevalence in the oldest age groups (amounting to more than 70% in patients aged over 50 years). Such prevalence is higher compared to the previous studies using the first-generation test and compared to the Spanish [17,19] and a multicentre Portuguese trial [20]. The data obtained from patients aged over 60 years are minimal. Regarding the age quintile, such a group was recently assessed only in the Finnish studies, in which, similarly to Poland, the JCV seropositivity rate reached almost 80% [18]. 

Compared to the European results, the data taken from East Asia are quite impressive. However, in China and South Korea, the relation mentioned above between the serostatus and patients’ age is not observed [15,16]. In those countries, the high anti-JCV Ab seroprevalence already occurs in the youngest age groups reaching a plateau level over time. This may suggest that, in those regions of the world, the latent JCV infection occurs much earlier compared to European countries.

The second demographic factor linked with a higher JCV seroprevalence, based on the results of the previous studies, was the male sex [12,13,18,23,25,26,30,33,36]. However, the results of our studies do not confirm the above relation, which is consistent with recently presented data concerning patients treated both in Europe and East Asia, where the first- [37] and second-generation tests were used [15,17,19,20,27].

We only have limited data derived from a few studies evaluating the potential link between the JCV prevalence and demographic and clinical factors, such as disease duration, disease course and disability measured on EDSS score. In the previous studies using the STRATIFY JCV™ test, no such correlation was identified [13,36]. Similarly, the use of the STRATIFY JCV™ DxSelect test did not allow for confirming the link between the serostatus and disability [16], disease duration [16,17,18,19,20] or the disease course [17,18]; however, the Spanish and Finnish cohorts of PPMS and SPMS patients were small. Initially, our results indicated the presence of such correlations. However, a further statistical analysis performed with the use of ANCOVA model showed that the patient’s age is the only factor related to an increase in anti-JCV Ab prevalence, which is consistent with the previous data.

The essential objective of this study was to assess the anti-JCV Ab serostatus in treatment-naïve patients and subjects receiving immunomodulatory and immunosuppressive agents because such data are scarce. The recent, multicentre Portuguese trial did not show any differences between PwMS receiving and not receiving immunosuppressive treatment; however, that study was limited by the very small group of patients treated with immunosuppressants [20]. Similarly, recent Spanish and Chinese studies did not confirm that the type of the treatment used impacts the JCV seroprevalence rate [16,19]; however, in this case, the groups of patients receiving immunosuppressants were also small. Such relations were observed in older studies, both in the Portuguese and Canadian cohorts forming the JEMS group [36,37], where patients receiving immunosuppressants obtained a significantly higher seroprevalence rate compared to the treatment-naive group, 63% vs. 55.9% [36]. A similar trend (but insignificant) was reported for the entire JEMS group [13].

Our study demonstrated markedly higher anti-JCV Ab seroprevalence in the group treated with immunosuppressants (76%) compared to a treatment-naïve (65.3%) group or patients receiving immunomodulatory treatment (62.4%). Importantly, the difference in the serostatus was statistically significant. The reported seroprevalence of anti-JCV Ab is much higher compared to the Canadian cohort [36]. However, compared to studies using the STRATIFY JCV™ DxSelect test to assess the serostatus, no such relation was observed; our group was much more numerous [16,19,20]. The assessment of individual classical immunosuppressants did not demonstrate any significant differences in the seroprevalence rates between the drugs. The lower seropositivity rate in patients receiving immunomodulatory or selective immunosuppressants compared to the treatment-naïve cohort, in which the majority of patients were treated with INFβ, potentially indicates the protective effect of such a therapy. Especially as up to date, no case of the PML was reported during the long-term use of IFNβ and glatiramer acetate [38,39]. The low anti-JCV Ab prevalence in patients treated with dimethyl fumarate (45.5%) compared to other immunomodulatory drugs is puzzling; however, the reported difference was insignificant. A potential reason for this situation could be a small group of patients receiving DMF; however, it requires assessment in further studies. The seroprevalence rate in the FGL and NTZ groups was impacted by the previously mentioned Polish criteria used for treatment qualification. Nevertheless, we decided to include those patients in our study in order not to skew the assessment of the entire study cohort.

Regarding the anti-JCV Ab levels, we only have data which were obtained in recent years. The mean AI level (1.41) for our entire study cohort as well as for seropositive patients (2.09) is similar to the results of the Czech patients (1.29 and 2.09, for the entire study group and seropositive patients, respectively) and with results obtained for seropositive Portuguese (2.1) and Iranian (2.23) patients [20,28,34]. Similarly, the median AI level reported in our cohort is consistent with the Austrian results (2.3), but markedly higher compared to the Finnish study (1.64) [18,31]. The highest AI levels in PwMS were observed, similarly to the seroprevalence rate, in East Asia countries, where the value obtained for the entire cohort amounted to 3.17 in China and 3.27 in South Korea [15,16].

Similarly to the JCV seroprevalence rate increasing with the patient’s age, our analysis also showed that the anti-JCV antibody index (AI) significantly increases in correlation with older age. The similar relation was observed in Austrian and Czech studies, while the presented trend concerned the German cohort [28,31,39]. Furthermore, no such relation, similarly to the serostatus rate, was observed in the South Korean study [15]. Similarly to the results from the previous multicentre Spanish and Czech and Austrian studies, we also did not demonstrate any correlation between the anti-JCV Ab index and the patient’s sex, disease duration or disease course [17,28,31].

There are virtually no existing data reporting this issue. Only the previous Austrian data show that prior use of disease-modifying therapy (DMT) does not impact the anti-JCV Ab index levels [31]. Our results on the AI levels distribution by the treatment used are in contrast with the above data. More interestingly, they are consistent with our data on the seroprevalence rate. In the entire study cohort, the highest index levels were observed in patients receiving immunosuppressants (1.67) followed by treatment-naïve (1.44) and immunomodulatory-treated (1.35) groups. Our results suggest why the previous immunosuppressive treatment is a risk factor for PML development, especially in patients subsequently treated with natalizumab. The results obtained for individual immunosuppressants are quite impressive, showing the highest AI levels in patients receiving classical immunosuppressive MS treatment, such as mitoxantrone, for which mean and median AI levels were 1.75 and 1.92, respectively. Further, worrying results were observed for cladribine, which is used in Poland in its parenteral form for many years in the treatment of multiple sclerosis [40]. Despite the lowest seroprevalence rate of the analysed immunosuppressants, the mean index level for CLA was 1.69; thus, it was only slightly lower compared to Mx and markedly higher vs. cyclophosphamide (mean AI: 1.50). It may be crucial in clinical practice because, after the completion of phase III trials, its oral form was approved for the treatment of RRMS [41]. Although no cases of PML were previously described in PwMS receiving cladribine, there are reports about such infections in patients receiving the drug for other indications [42].

We observed the analogical situation concerning the consistency of serostatus and AI levels results in patients receiving immunomodulatory treatment. In this case, AI levels, similarly to the seroprevalence rate, were also lower compared to the treatment-naïve group. Differences in the anti-JCV AI levels between individual immunomodulatory drugs were not significant. However, similarly to the anti-JCV Ab seroprevalence rate, the lowest mean AI level (1.11) was reported in the DMF group. Taking into account a few PML cases reported in DMF-treated PwMS with coexistent leukopenia [38], one can hypothesise that leukopenia itself is the main factor for PML development. However, it warrants further, more extensive studies, especially longitudinal studies, to determine the anti-JCV AI levels before and during DMF treatment. We should also check whether long-term DMF use, similarly to rituximab, correlates with a progressive decrease in AI levels. On the one hand, the AI levels measured during treatment with rituximab are distorted and do not reflect the actual values, while, on the other hand, it is known that rituximab used in other than MS indications is responsible for the occurrence of 1 PML case per 25,000 treated patients [43,44].

Only a few studies provide data about the categories of the anti-JCV Ab index. Previously, no extensive analysis of the AI categories in correlation with the demographic and clinical factors had been performed. Moreover, the actual results come from different populations of MS patients, thereby making any comparative analysis difficult.

It seems that the high-risk category is the most important. Our analysis of the prevalence of AI categories showed that patients with AI levels > 1.5 form the biggest group of PwMS, i.e., 41.6% of the entire study cohort. When compared with the results of the previous European studies, our high-risk category cohort is more numerous than the Spanish (39.9% and 35.4% in natalizumab- or immunosuppressant-naive groups, respectively), British (33%), French (33%) and German (22%) groups. However, it is less numerous compared to the Austrian (45%) and Portugal immunosuppressant-naive cohorts (45.5%), which to date were an example of the highest prevalence of the high-risk AI category in European countries [17,19,20,27,29,31,39].

However, the prevalence of AI levels > 1.5 (63.1%) in our entire cohort of seropositive patients is almost the same as in the Portuguese immunosuppressant-naive group (63.7%) and higher compared to the other Portuguese group (59.2%) [20,30]. Similarly to the seroprevalence rate and anti-JCV Ab index levels, the highest prevalence of the high-risk index category was observed in East Asia countries—from 52.4% (Japan) to 61% (both in China and South Korea) in the entire cohort, and from 75% to 78% in the seropositive group [14,15,16].

Moreover, our analysis showed that AI levels > 3.0 are observed even in 28.1% of seropositive patients. For comparison, the only existing South Korean data revealed that such a high index category is observed in up to 56% of patients [15]. This confirms the presence of the apparent differences between Asian and European populations and demonstrates the higher risk of PML occurrence in the former cohort.

The assessment of the correlation between the individual anti-JCV Ab index categories and the demographic or clinical factors did not show any statistically significant differences. However, the analysis of AI categories distribution by age quintiles demonstrates a reduction in values in the seronegative and undetermined categories occurring along with increasing age, and a progressive increase of value in the seropositive categories. In this case, the difference was also insignificant. The above results are consistent with the data on the seroprevalence rates and AI levels depending on the analysed demographic and clinical factors.

Similarly to the demographic and clinical factors described above, the previous studies also did not include any analysis of the impact of DMT used on the distribution of anti-JCV Ab index categories. We observed a significant distribution of values within high-risk category (i.e., >1.5) for the entire study cohort, with the lowest and the highest prevalence in patients treated with immunomodulatory (38.3%) and immunosuppressive agents (51.4%), respectively. The presented results are entirely consistent with data concerning the seroprevalence rate and AI levels and confirm and the critical role of immunosuppression as a risk factor for PML development. The further detailed assessment of categories distribution by MS drug used shows that, in patients receiving mitoxantrone or cladribine (AI > 1.5 in 58.2% and 46.4% of patients, respectively), we should consider anti-JCV Ab index monitoring; especially in case of treatment failure or a planned therapy switch to one of the monoclonal antibodies (mAb) used in MS treatment. Regarding immunomodulatory drugs, the analysis of index categories showed that, in a high-risk group, the lowest prevalence concerns patients receiving DMF; however, compared to IFN and GA, the difference was insignificant. This is consistent with the data concerning AI values.

The prevalence of anti-JCV AI > 1.5 observed in the entire cohort (40%) and PwMS receiving immunosuppressive treatment (50%) in our study has crucial importance for the selection of MS therapy in clinical practice; especially in the era of new highly-effective drugs. At present, we know that PML cases occur not only in MS patients receiving natalizumab but also in subjects treated with fingolimod, dimethyl fumarate or ocrelizumab as well as in patients receiving rituximab or cladribine due to other indications [7,11,38,42]. Taking that into account, we need to conduct further studies evaluating the long-term changes in seroprevalence rates and the antibody index with the potential analysis of blood parameters, such as leukocytes and lymphocyte subpopulations levels. This could be important as the example of rituximab, or dimethyl fumarate, shows that an evaluation of anti-JCV Ab index levels could be inadequate. Our data suggest that immunosuppressant-treated patients qualifying for therapy with highly-effective, modern MS therapies, regardless of the drug considered in the therapeutic scheme, should undergo tests to identify the potential presence and to measure the level of the anti-JCV Ab index.

Our study has some limitations, including a small group of patients receiving fingolimod and natalizumab as well as dimethyl fumarate. The specific criteria used to qualify patients for treatment with the first two agents impacted the analysis, in which the data derived from those patients were only assessed as components of the entire cohort. There is also the lack of longitudinal data, especially important information for DMF and cladribine.

## 5. Conclusions

In our multicentre, prospective study involving 1405 Polish MS patients, the presence of anti-JCV Ab was observed in 65.8% of the participants—one of the highest records of JC virus seroprevalence in Europe. Of the demographic factors, only the patient’s age positively correlated with seroprevalence. Our data indicated that patients undergoing immunosuppressive treatment achieved markedly higher anti-JCV Ab prevalence, had higher AI levels, and formed the largest group of patients with high-risk AI level category compared to treatment-naive and immunomodulatory-treated cohorts. On the one hand, we have observed high anti-JCV Ab prevalence and AI levels in the cladribine group, while, on the other hand, those parameters in the dimethyl fumarate group were low. Therefore, there is a need to conduct longitudinal studies to address this issue and to determine the actual and future importance of treatment individualisation in the future, especially during qualification to treatment with highly-effective agents.

## Figures and Tables

**Figure 1 jcm-10-01998-f001:**
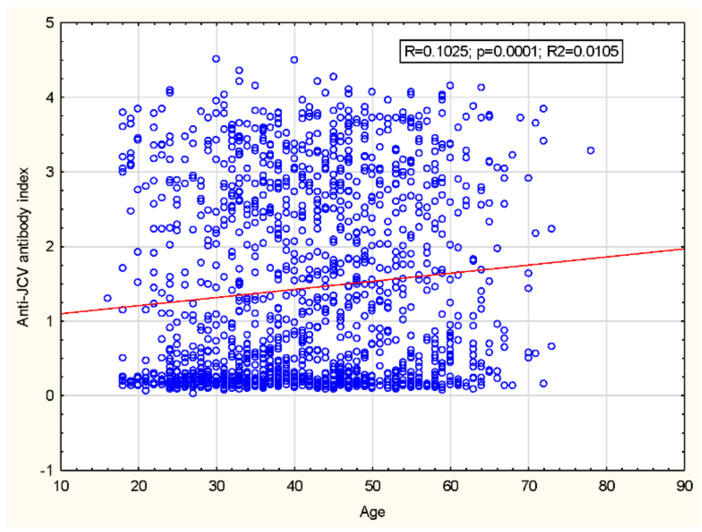
Correlation between anti-JCV Ab index and patient’s age. r = Spearman = 0.1025; *p* = 0.0001.

**Figure 2 jcm-10-01998-f002:**
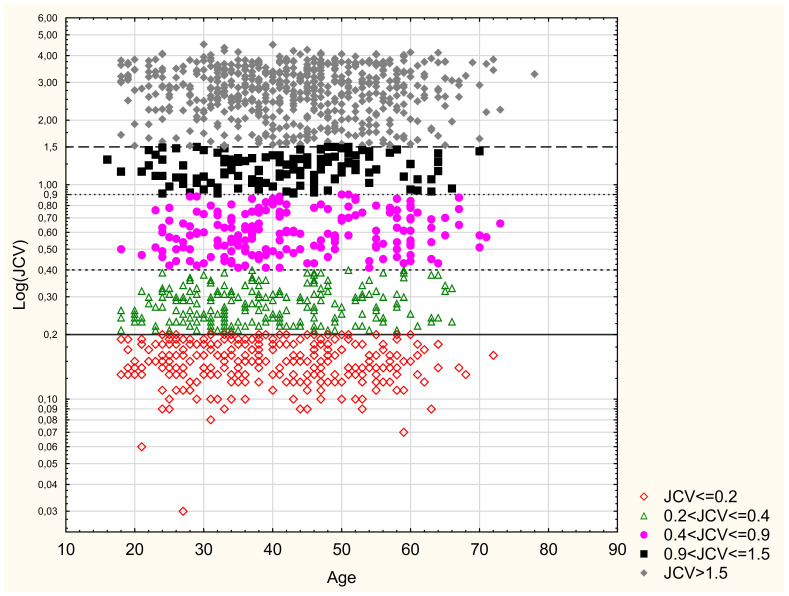
Anti-JCV AI categories and age.

**Table 1 jcm-10-01998-t001:** Clinical and demographic characteristics of MS patients at the time of the STRATIFY JCV DxSelect test.

Characteristic	Study Population
Number of patients, *n*	1405
Sex, female/male, *n*	990/415
Mean age, years ± SD	41.7 ± 12.2
Age, median (IQR; range)	41.0 (19.0; 18–78)
Age categories, *n* (%)	
18–29 years	248 (17.7)
30–39 years	396 (28.2)
40–49 years	381 (27.1)
50–59 years	267 (19.0)
≥60 years	113 (8.0)
Disease course, *n* (%)	
RRMS	955 (68.0)
SPMS	268 (19.0)
PPMS	182 (13.0)
Disease duration, mean (SD); median (IQR; range)	8.5 (8.2); 6.0 (10.0; 0–50)
Disease duration categories, *n* (%)	
0–5 years	663 (47.2)
6–10 years	311 (22.1)
11–15 years	194 (13.8)
≥16 years	237 (16.9)
EDSS, mean (SD); median (IQR; range)	3.5 (1.9); 3.0 (3.0; 1.0–9.0)
Previous and actual treatment, *n* (%)	
Treatment naïve	665 (47.3)
Immunomodulation and selective immunosuppression	532 (37.9)
IFN-β	367 (26.1)
GA	85 (6.0)
DMF	22 (1.6)
FGL	33 (2.4)
NTZ	25 (1.8)
Immunosuppression	208 (14.8)
Mx	115 (8.2)
CTX	52 (3.7)
CLA	41 (2.9)

RRMS—relapsing–remitting multiple sclerosis, SPMS—secondary progressive multiple sclerosis, PPMS—primary progressive multiple sclerosis, SD—standard deviation, IFN—interferon, GA—glatiramer acetate, DMF—dimethyl fumarate, FGL—fingolimod, NTZ—natalizumab, Mx—mitoxantrone, CTX—cyclophosphamide, and CLA—cladribine in vials for subcutaneous administration.

**Table 2 jcm-10-01998-t002:** Anti-JCV Ab prevalence and antibody index by patients’ sex, disease course and EDSS score.

*n* (%)	All	F	M	MW-U	RRMS	SPMS	PPMS	KW-H Post hoc	EDSS I	EDSS II	EDSS III	KW-H Post hoc
JCV+	924 (65.8)	654 (66.1)	270 (65.1)	*p* > 0.05	604 (63.2)	183 (68.3)	137 (75.3)	KW-H *p* = 0.005 RRMS-PPMS *p* = 0.03 Other *p* > 0.05	313 (60.9)	343 (67.1)	268 (70.5)	KW-H *p* = 0.008 EDSS(1)-EDSS(3) *p* = 0.04 Other *p* > 0.05
JCV−	481 (34.2)	336 (33.9)	145 (34.9)		351 (36.8)	85 (31.7)	45 (24.7)		201 (39.1)	168 (32.9)	112 (29.5)	
AI Mean	1.44	1.43	1.47	*p* > 0.05	1.39	1.46	1.71	KW-H *p* = 0.009 RRMS-PPMS *p* = 0.006 Other *p* > 0.05	1.35	1.46	1.55	KW-H *p* = 0.06
AI Median	0.95	0.96	0.93		0.79	1.04	1.75		0.72	1.02	1.19	
AI Min	0.03	0.03	0.09		0.03	0.09	0.09		0.06	0.03	0.07	
AI Max	4.51	4.51	4.50		4.51	4.07	4.13		4.51	4.50	4.13	
AI SD	1.29	1.28	1.32		1.29	1.29	1.28		1.29	1.29	1.29	

F—female, M—male, RRMS—relapsing–remitting multiple sclerosis, SPMS—secondary progressive multiple sclerosis, PPMS—primary progressive multiple sclerosis, EDSS I—from 0 to 2.0, EDSS II—from 2.5 to 4.5, EDSS III—from 5.0 to 9.0, AI—antibodies index, and SD—standard deviation.

**Table 3 jcm-10-01998-t003:** Anti-JCV Ab prevalence and antibody index by patients’ age and disease duration (years).

	Age Categories						Disease Duration Categories				
*n* (%)	18–29	30–39	40–49	50–59	≥60	KW-H Post hoc	0–5	6–10	11–15	≥16	
JCV+, *n* (%)	130 (52.4)	254 (64.1)	261 (68.5)	189 (70.8)	90 (79.6)	KW-H *p* < 0.0001; 18–29–40–49 *p* = 0.006; 18–29–50–59 *p* = 0.003; 18–29–≥60 *p* = 0.0003; Other *p* > 0.05	413 (62.3)	206 (66.2)	135 (69.6)	170 (71.7)	KW-H *p* = 0.03; 0–5–11–15 *p* = 0.06; 0–5–≥16 *p* = 0.009 Other *p* > 0.05
JCV−, *n* (%)	118 (47.6)	142 (35.9)	120 (31.5)	78 (29.2)	23 (20.4)		250 (37.7)	105 (33.8)	59 (30.4)	67 (28.3)	
AI Mean	1.15	1.40	1.56	1.58	1.55	KW-H *p* = 0.0001; 18–29–30–39 *p* = 0.06; 18–29–40–49 *p* = 0.0002 18–29–50–59 *p* = 0.001; 18–29–≥60 *p* = 0.003; Other *p* > 0.05	1.38	1.45	1.44	1.61	KW-H *p* > 0.05
AI Median	0.44	0.75	1.28	1.37	0.96		0.74	1.11	0.89	1.40	
AI Min	0.03	0.08	0.09	0.07	0.09		0.06	0.03	0.09	0.09	
AI Max	4.09	4.51	4.50	4.07	4.15		4.27	4.51	4.50	4.21	
AI SD	1.24	1.29	1.29	1.30	1.27		1.30	1.28	1.27	1.30	

AI—antibodies index and SD—standard deviation.

**Table 4 jcm-10-01998-t004:** Anti-JCV Ab prevalence and antibody index by patients’ treatment.

*n* (%)	Naive	IS	IM	KW-H	CLA	CTX	Mx	KW-H	IFN	GA	DMF	FGL	NTZ	KW-H
JCV+	434 (65.3)	158 (76.0)	332 (62.4)	KW-H *p* = 0.002; MW-UIS/IM: *p* = 0.01 Other *p* > 0.05	30 (73.2)	41 (78.8)	87 (75.7)	KW-H *p* > 0.05	234 (63.8)	58 (68.2)	10 (45.5)	27 (81.8)	3 (12.0)	KW-H *p* = 0.002; *p* < 0.05 only for NTZ comparison
JCV−	231 (34.7)	50 (24.0)	200 (37.6)		11 (26.8)	11 (21.2)	28 (24.3)		133 (36.2)	27 (31.8)	12 (54.5)	6 (18.2)	22 (88.0)	
AI Mean	1.45	1.67	1.36	KW-H *p* = 0.004 IS/IM: *p* = 0.0008 MW-U Naive/IS: MW-U *p* = 0.01 Naive/IM: *p* = 0.02	1.69	1.50	1.75	KW-H *p* > 0.05	1.37	1.47	1.11	1.75	0.51	KW-H *p* < 0.0001 NTZ-GA *p* = 0.00002; NTZ-IFN *p* = 0.00002; NTZ-S1P *p* = 0.000008; other *p* > 0.05
AI Median	0.93	1.55	0.74		1.16	1.26	1.92		0.75	1.14	0.30	1.55	0.16	
AI Min	0.03	0.09	0.06		0.11	0.10	0.09		0.06	0.09	0.08	0.13	0.10	
AI Max	4.36	4.13	4.51		4.13	3.82	4.04		4.51	3.78	4.03	4.05	3.72	
AI SD	1.30	1.25	1.28		1.42	1.16	1.23		1.29	1.27	1.35	1.18	0.95	

IS—classical immunosuppressants, IM—immunomodulatory drugs and selective immunosuppressants, IFN—interferon, GA—glatiramer acetate, DMF—dimethyl fumarate, FGL—fingolimod, NTZ—natalizumab, Mx—mitoxantrone, CTX—cyclophosphamide, CLA—cladribine in vials for subcutaneous administration, AI—antibodies index, and SD—standard deviation.

**Table 5 jcm-10-01998-t005:** Anti-JCV AI categories and sex, clinical course and EDSS score.

*n* (%)	All	F	M	RRMS	SPMS	PPMS	EDSS I	EDSS II	EDSS III
JCV AI ≤ 0.2	291 (20.7)	205 (20.7)	86 (20.7)	204 (21.4)	57 (21.3)	30 (16.5)	110 (21.4)	108 (21.1)	73 (19.2)
0.2 < JCV AI ≤ 0.4	212 (15.1)	148 (14.9)	64 (15.4)	158 (16.5)	37 (13.8)	17 (9.3)	100 (19.5)	67 (13.1)	45 (11.8)
0.4 < JCV AI ≤ 0.9	188 (13.4)	131 (13.2)	57 (13.7)	132 (13.8)	34 (12.7)	22 (12.1)	65 (12.6)	70 (13.7)	53 (13.9)
0.9 < JCV ≤ AI1.5	129 (9.2)	96 (9.7)	33 (8.0)	84 (8.8)	26 (9.7)	19 (10.4)	45 (8.8)	48 (9.4)	36 (9.5)
JCV AI > 1.5	585 (41.6)	410 (41.4)	175 (42.2)	377 (39.5)	114 (42.5)	94 (51.6)	194 (37.7)	218 (42.7)	173 (45.5)
		MW-U *p* > 0.05		KW-H *p* > 0.05			KW-H *p* > 0.05		

F—female, M—male, RRMS—relapsing–remitting multiple sclerosis, SPMS—secondary progressive multiple sclerosis, PPMS—primary progressive multiple sclerosis, EDSS I—from 0 to 2.0, EDSS II—from 2.5 to 4.5, EDSS III—from 5.0 to 9.0, and JCV AI—anti-JCV Ab index.

**Table 6 jcm-10-01998-t006:** Anti-JCV AI categories and patient’s age and disease duration.

	Age Categories					Disease Duration Categories			
*n* (%)	18–29	30–39	40–49	50–59	≥60	0–5	6–10	11–15	≥16
JCV AI ≤ 0.2	71 (28.6)	82 (20.7)	70 (18.4)	54 (20.2)	14 (12.4)	151 (22.8)	62 (19.9)	31 (16.0)	47 (19.8)
0.2 < JCV AI ≤ 0.4	50 (20.2)	69 (17.4)	52 (13.6)	30 (11.2)	11 (9.7)	110 (16.6)	45 (14.5)	33 (17.0)	24 (10.1)
0.4 < JCV AI ≤ 0.9	29 (11.7)	55 (13.9)	40 (10.5)	36 (13.5)	28 (24.8)	83 (12.5)	39 (12.5)	35 (18.0)	31 (13.1)
0.9 < JCV AI ≤ 1.5	22 (8.9)	33 (8.3)	42 (11.0)	21 (7.9)	11 (9.7)	59 (8.9)	33 (10.6)	16 (8.2)	21 (8.9)
JCV AI > 1.5	76 (30.6)	157 (39.6)	177 (46.5)	126 (47.2)	49 (43.4)	260 (39.2)	132 (42.4)	79 (40.7)	114 (48.1)
KW-H	*p* > 0.05					*p* > 0.05			

JCV AI—anti-JCV Ab index.

**Table 7 jcm-10-01998-t007:** Anti-JCV AI categories and patient’s treatment.

*n* (%)	Naïve	IS	IM	KW-H	CLA	CTX	Mx	KW-H	IFN	GA	DMF	FGL	NTZ	KW-H
JCV AI ≤ 0.2	142 (21.4)	30 (14.4)	119 (22.4)	KW-H *p* = 0.0007 post hoc IS/IM *p* = 0.0009 IS/Naïve *p* = 0.01 IM/Naïve *p* > 0.05	6 (14.6)	9 (17.3)	15 (13.0)	KW-H *p* > 0.05	73 (19.9)	16 (18.8)	8 (36.4)	5 (15.2)	17 (68.0)	KW-H *p* < 0.00001 post hoc GA/NTZ *p* = 0.00006 IFN/NTZ *p* = 0.0005 S1P/NTZ *p* = 0.00001 Other *p* > 0.05
0.2 < JCV AI ≤ 0.4	99 (14.9)	23 (11.1)	90 (16.9)		5 (12.2)	4 (7.7)	14 (12.2)		66 (18.0)	13 (15.3)	4 (18.2)	2 (6.1)	5 (20.0)	
0.4 < JCV AI ≤ 0.9	87 (13.1)	25 (12.0)	76 (14.3)		8 (19.5)	6 (11.5)	11 (9.6)		60 (16.3)	11 (12.9)	3 (13.6)	2 (6.1)	–	
0.9 < JCV AI ≤ 1.5	63 (9.5)	23 (11.1)	43 (8.1)		3 (7.3)	12 (23.1)	8 (7.0)		27 (7.4)	9 (10.6)	1 (4.5)	6 (18.2)	–	
JCV AI > 1.5	274 (41.2)	107 (51.4)	204 (38.3)		19 (46.3)	21 (40.4)	67 (58.3)		141 (38.4)	36 (42.4)	6 (27.3)	18 (54.5)	3 (12.0)	

IS—classical immunosuppressants, IM—immunomodulatory drugs and selective immunosuppressants, IFN—interferon, GA—glatiramer acetate, DMF—dimethyl fumarate, FGL—fingolimod, NTZ—natalizumab, Mx—mitoxantrone, CTX—cyclophosphamide, CLA—cladribine in vials for subcutaneous administration, and JCV AI—anti-JCV Ab index.

## Data Availability

The data presented in this study are available on request from the corresponding author. The data are not publicly available due to privacy restrictions.

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
