# Peer review of "JC Virus Seroprevalence and JCVAb Index in Polish Multiple Sclerosis Patients Treated with Immunomodulating or Immunosuppressive Therapies"

_jcm, 2021, doi:10.3390/jcm10091998_

Round 1
Reviewer 1 Report
The authors have conducted a multi-site study in which serum samples have been collected from MS patients and tested at a single reference laboratory for JCV antibody. The test method used is widely accepted as a standard for JCV antibody status. The results have been used to determine JCV seroprevalence and analyses have been performed to assess various associations with treatments undertaken. Determination of JCV status is usually considered a prerequisite prior to treatment with natalizumab. This is to reduce the risk of PML due to JCV reactivation. JCV seroprevalence of c. 50% has been widely documented in adults and it is known to increase with age. This study makes similar findings.
For me, there were too many tables and the data was difficult to analyse. Table 1 in my version of the paper was illegible. It would be helpful if 95% confidence intervals were included to help inform the reliability of the observations. I found it hard to accept increasing seroprevalences with treatment type, as seroprevalence is primarily governed by the rate of infection particularly during childhood. A scatterplot of activity indices by treatment type would be useful. Overall, a comprehensive piece of work; however, I found it hard work to follow and I did not find the tables helpful.
In the text, I found the use of . instead of , to indicate "thousand" (eg. line 28) to be strange.
Author Response
Dear Reviewer,
Thank you for your valuable comments and feedback regarding our research. We have made some changes to the manuscript, as well as recorded the answers to the comments below.
For me, there were too many tables and the data was difficult to analyse. Table 1 in my version of the paper was illegible. It would be helpful if 95% confidence intervals were included to help inform the reliability of the observations. I found it hard to accept increasing seroprevalences with treatment type, as seroprevalence is primarily governed by the rate of infection particularly during childhood. A scatterplot of activity indices by treatment type would be useful. Overall, a comprehensive piece of work; however, I found it hard work to follow and I did not find the tables helpful.
Table 1 is corrected (error in converting the paper).
Some corrections were made to the tables to improve their legibility.
As Mann Whitney U Test and Kruskal Wallis H test were used for statistical analysis, it is not required to include 95% CI. Moreover, the tables would be illegible if additional data were included.
JCV seroprevalence is related to the rate of infection during childhood. However, seroprevalence increases with age and is associated with the virus genotype (included in the Discussion section). In the previous studies, conducted on smaller groups, the association between treatment type and JCV seroprevalence was noticed as well (discussed in the paper).
A scatterplot of AI by treatment is attached (please see the attachment). However, we did not add it to the manuscript, as it do not present the results better than the table in our opinion.
In the text, I found the use of . instead of , to indicate "thousand" (eg. line 28) to be strange.
It is corrected.

Reviewer 2 Report
In this multi-institutional study of 1405 MS patients from Poland, the authors report the findings of JCV Ab index testing. Perhaps the most novel finding from the study is the difference in JCV Ab index results in individuals on immunomodulating versus immunosuppressive agents. However, it is difficult to interpret precisely what this means as the authors failed to indicate when the antibody testing was performed relative to the start of the drug. Typically, at least in the U.S., this test is performed prior to the institution of a DMT as it helps guide the selection of the therapy. If this were the case in this study, then the results are irrelevant.
The JCV Ab test was repeated in only 58 patients (on fingolimod and natalizumab) and making it difficult to state what the effects of the DMTs were on JCV antibody index levels.
In the same vein, categorizing disease modifying therapies (DMTs) into immunomodulatory versus immunosuppressive agents is thorny and controversial. The authors should address why they characterize the drugs as they did.
The manuscript is excessively long and needs to be considerably shortened.
The comment (line 56) that infection is "self-limiting" should be changed to "inapparent" and in that same sentence, the comment that the "viral load becomes undetectable" would need a reference. What is the study that supports that?
In the paragraph starting at line 77, the authors imply that JCV Ab index may be applicable to PML developing with other DMTs apart from natalizumab; however, the frequency with which it is seen with other DMTs is so low to preclude a meaningful analysis of whether it can be used as a risk mitigation strategy for these agents. Given the low frequency with these DMTs, it would almost certainly not be worthwhile. Some comment regarding this aspect of JCV Ab index measurement with DMTs other than natalizumab and perhaps DMF where there is consensus that both JCV Ab seropositivity and low lymphocyte counts warrant its discontinuation would be valuable.
As the authors note and has been amply demonstrated in previous studies, seropositivity rates correlate with advancing age and this is responsible for the apparent association with progressive MS and EDSS values.
While there may be differences between the seropositivity rates among various European populations, without controlling for age in the different populations studied, it is not a particularly valuable observation.
Round 2
Reviewer 2 Report
The authors have addressed this reviewer's concerns.